



# A principal component based strategy for regionalisation of precipitation intensity-duration-frequency (IDF) statistics

Kajsa M. Parding[1], Rasmus E. Benestad[1], Anita Verpe Dyrrdal[1], and Julia Lutz[1]

[1]Norwegian Meteorological Institute

**Correspondence:** Kajsa M. Parding (kajsa.parding@met.no)

**Abstract.** Intensity-duration-frequency (IDF) statistics describing extreme rainfall intensities in Norway were analysed with the purpose of investigating how the shape of the curves is influenced by geographical conditions and local climate characteristics. To this end, principal component analysis (PCA) was used to quantify salient information about the IDF curves and a Bayesian linear regression was used to study the dependency of the shapes on climatological and geographical information.

Our analysis indicated that the shapes of IDF curves in Norway are influenced by both geographical conditions and 24-hr precipitation statistics. Based on this analysis, an empirical model was constructed to predict IDF curves in locations with insufficient sub-hourly rain gauge data and for the future using downscaled data from global climate models. Our new method was also compared with a recently proposed formula for estimating sub-daily rainfall intensity based on 24-hr rain gauge data. We found that a Bayesian inference of a PCA representation of IDF curves provides a promising strategy for estimating

sub-daily return levels for rainfall.

## 1 Introduction

Climate change caused by an increased greenhouse effect is expected to be associated with changes in the hydrological cycle and an increase in precipitation and more extreme rainfall (Field et al., 2012; Stocker and Qin, 2013; Solomon et al., 2007;

IPCC, 2021). There are several physics-based explanations for the increased rainfall amounts. Higher surface temperature gives higher rates of evaporation and strengthens the moisture-holding capacity of air. The air moisture is part of the global hydrological cycle where it condenses to form clouds and returns to Earth's surface through precipitation. Furthermore, a recent analysis of the satellite-based Tropical Rainfall Measuring Mission (TRMM) suggests that there has been a change in the global rainfall area over the recent decades that may also imply changes in the mean rainfall intensity (Benestad, 2018) and

there are similar indications in recent state-of-the-art reanalyses (Benestad et al., 2022). There is also a possibility that some rain-producing phenomena become more prevalent under warmer conditions, such as convective systems. These theoretical aspects are underscored by trend analyses of the probability of heavy rainfall pointing to more extreme rainfall amounts (Benestad et al., 2019b, 2021; Westra et al., 2014; Donat et al., 2016; Sorteberg et al., 2018; Dyrrdal et al., 2021; Olsson et al.,





2022). Extreme rainfall is a disruptive and damaging hazard but can to some extent be managed through a proper risk analysis.

For example, precipitation intensity-duration-frequency (IDF) curves are commonly used tools in water resource management and planning (Koutsoyiannis et al., 1998; Mailhot et al., 2007; Burn, 2014; Dyrrdal et al., 2015).

One problem associated with IDF curves is that rain gauge data have limited geographical coverage, making it difficult to analyse the risk of extreme rainfall everywhere. The Intergovernmental Panel on Climate Change's sixth assessment report also noted that the exact levels of regional IDF characteristics may depend on the method as well as the resolution of downscal-

ing when derived from climate model simulations (IPCC, 2021). Furthermore, short time series imply limited knowledge of extremes and challenges the extrapolation to long return periods. These caveats are particularly problematic for IDF statistics for short-duration rainfall (3 hours or less), which are useful for the design of urban infrastructure and urban flood prevention. Short-duration rainfall is often a result of local convective activity and hence highly variable, but sub-hourly precipitation observations are relatively sparse. One way to circumvent this issue is by using systematic dependencies between IDF curves and

climatological and geographical factors to regionalise the IDF curves from gauged to ungauged stations. While each precipitation event typically has low predictability, the statistical nature of such events, such as their probability, frequency or typical intensity, may still be highly predictable. Dyrrdal et al. (2015) developed a method for estimating return levels for sub-daily rainfall intensity over Norway based on high-resolution gridded climate observations and Bayesian hierarchical modelling. One caveat with using gridded data is that it is not optimal for extreme values due to spatial inhomogeneity resulting from

the way the data is sampled (Schilcher et al. (2017); Richard Chandler, private communications). It is also impractical and computationally expensive to calculate IDF curves for each grid point. Using a statistical model that utilises geographical and meteorological information from gridded products, but is tuned to local data, may be a more efficient approach. A similar principle may in addition be used to expand IDF curves from the current period (gauged or ungauged) to the future and hence provide added value in terms of increased versatility. Another problem is that IDF curves may be inconsistent across durations.

Roksvåg et al. (2021) proposed two post-processing approaches to deal with such inconsistencies. Another approach may involve a sort of 'temporal downscaling' of sub-daily rainfall intensity based on 24-hr precipitation statistics (Benestad et al., 2021). In a similar vein, Rodríguez et al. (2014) estimated future hourly extreme rainfall using temporal downscaling based on scaling properties of rainfall for a case study of Barcelona and the SRES A1B, A2 and B2 climate scenarios. Fauer et al. (2021) used a different approach, proposing a flexible and consistent quantile estimation method for IDF curves that explored how

to improve the estimation by adjusting parameters representing curvature, multi-scaling and flatness. Their approach involved using a duration-dependent formulation of the generalised extreme value (GEV) distribution to fit IDF models with a range of durations simultaneously. The degree of uncertainty associated with estimating IDF curves is substantial, and Chandra et al. (2015) attempted to quantify uncertainties connected to both insufficient quantity and quality of data, leading to parameter uncertainty due to the distribution fitted to the data, and uncertainty as a result of using multiple global climate models (GCMs)

from the CMIP5 ensemble (RCP 2.6, 4.5, 6.0 and 8.5 scenarios). They used a Bayesian approach and a case study of Bangalore city in India and found that the uncertainty is larger for shorter than for longer durations for the rainfall return levels, and also that parameter uncertainty was greater than the model uncertainty.





In this study, we test an empirical statistical modelling approach to estimate the shape of the IDF curves, rather than the
return values for each duration and return period. This approach is based on Principal Component Analysis (PCA) and Bayesian
inference. Using PCA reduced the return values of all stations and return periods to a set of principal components (PCs) in the
form of spatial patterns, with accompanying IDF shapes and eigenvalues. The leading PC spatial patterns were subjected to
Bayesian linear regression and subsequently expanded to new stations based on climatological and geographical information.
The analysis is included in its entirety in the R-markdown document provided in the Supplement. Predicting the shape of
the IDF curves of all return periods simultaneously through PCA is a novel strategy which to our knowledge has not been
done before in this context. The motivation for this approach was the observation and expectation that the curves have simple
and smooth shapes with regional similarities. In other words, the return values for rainfall intensities over different durations
are related to each other, and the IDF curves are associated with a substantial degree of redundant information that can be
utilised through the application of PCA. The estimated return values were compared with the simple formula for estimating
approximate values of return levels for sub-daily rainfall based on 24-hour rain gauge data presented in Benestad et al. (2021).
A strategy that involved using a weighted set of polynomials to represent the IDF shapes was also pursued but abandoned as it
provided a poor representation of the return values. The purpose of the statistical modelling was i) to explore the influence of
meteorological and geographical conditions on the IDF curves; ii) to establish an empirical model to be used for regionalisation
where sub-daily precipitation data are not available, iii) for statistical downscaling purposes, to be used in combination with
future projections of meteorological quantities to estimate changes in return values, and iv) compare different and independent
strategies for estimating sub-daily rainfall intensity.

## 2   Materials and Methods

### 2.1   Data

We used new IDF statistics from 74 Norwegian stations, consisting of return values for a range of durations (1, 2, 3, 5, 10,
15, 30, and 45 minutes, and 1, 1.5, 2, 3, 6, 12, and 24 hours) and return intervals (2, 5, 10, 20, 25, 50, 100, and 200 years),
depicted in Figure 1. The return values were calculated using the method described in Lutz et al. (2020) and post-processed with
the quantile selection algorithm from Roksvåg et al. (2021). The Roksvåg et al. post-processing was only applied to stations
where the IDF curves were not consistent (21 out of the stations). In the calibration, statistical properties based on rainfall
and temperature observations were used to represent local climatological conditions in addition to coordinates, elevation and
distance to the ocean. In this study, only stations where the IDF statistics were calculated based on at least 10 seasons of
precipitation data were considered. Eight stations with IDF statistics based on long precipitation records from different parts
of Norway were selected to display results visually (Figure 2). The IDF curves for all return periods for these stations are
displayed in Figure S4 of the Supplementary material.

Daily mean precipitation ($pr$) and air temperature at 2m ($t2m$) data were downloaded from the Norwegian Meteorological
Institute (https://frost.met.no), from the same or closest available stations as the IDF data. Only stations with at least 10 years
of data within the period 1970-2020 were included in the analysis. All IDF stations had daily precipitation data, but only 17 of





the 74 stations had available temperature data. At the stations that did not, the distance to the nearest station with temperature data was on average 6.3 km and at most 25 km.

## 2.2 Principal Component Analysis of the IDF curves

Principal component analysis (PCA) was applied to the IDF curves through Singular Value Decomposition (SVD) (Jolliffe, 1986; Jolliffe and Cadima, 2016; Trefethen and Bau, 1997). This framework was based on the expression

$$X = U\Lambda V^T, \tag{1}$$

where the matrix $X$ contains the IDF curves, i.e. return values for various return periods and durations at each location, $U$ is a matrix holding eigenvectors which can be interpreted as shapes of IDF curves, $\Lambda$ is a diagonal matrix holding the eigenvalues, and $V$ represents the principal components (PCs) containing weights for the different geographical locations.

The purpose of the PCA was to reduce the complexity of the IDF data while preserving as much variability as possible. The procedure finds new variables that are linear functions of the original data, where the new variables (PCs) are uncorrelated with each other and the variance is successively maximised, meaning that the leading PC describes the dominant pattern and each successive mode represents a smaller and smaller portion of the variance. The original IDF curves can thus be reproduced by combining a few of the leading PCs, eigenvectors and eigenvalues with little loss of information.

We consider several criteria for assessing the number $m$ of PCs that needs to be retained in order to represent most of the information:

i) Set $m$ to the smallest value for which the total cumulative explained variance exceeds a given percentage, e.g. 80% or 90%. (The explained variance can be calculated from the eigenvalues $\lambda$.)

ii) Using a Scree diagram where the eigenvalues are plotted against their rank, identify an 'elbow' (a bend in the curve where the slope goes from steep to shallow) and retain the eigenvalues to the left of this point (Cattell, 1966).

iii) An approach suggested by Ali et al. (1985) is to calculate the correlation between the original variables and the PCs and set $m$ to one less than the first PC for which there are no correlation coefficients that are statistically significantly different from zero. In our case, the "original variables" are the return values, which for each station was compared with the IDF shapes $U_i$ for i ∈ [1,10] (Equation 1).

There are other more objective but also computationally demanding methods of selecting the number of PCs to retain, e.g. based on cross-validation or bootstrapping, but these were not deemed necessary for our purposes. Based on the criteria above, we focused primarily only on the two leading PCs in most of the analysis and statistical modelling of this study (see discussion in Section 3.1).

## 2.3 Statistical modelling

Statistical relationships were established between the two leading PCs of the IDF curves, which represent the dominant spatial patterns of the data, and a set of geographical and meteorological predictors: the wet-day mean precipitation in the warm





season (April - September) and cold season (October - March), $\mu_{warm}$ and $\mu_{cold}$, the wet-day frequency in the warm season and cold season, $f_{w_{warm}}$ and $f_{w_{cold}}$, the temperature in the summer season (June - August), $t2m_{JJA}$, as well as the latitude, altitude, and minimum distance to the coast ($d_{ocean}$). These predictors describe both the cold season precipitation regime in

Norway, primarily dominated by stratiform precipitation associated with low-pressure systems, and the warm season precipitation regime, which to a larger degree is regulated by convective processes. The summer temperature was included because it is closely linked to the convective environment.

The statistical model can be described as follows:

$$\hat{V}_i(lon, lat) = c_{0,i} + c_{1,i}p_1(lon, lat) + c_{2,i}p_2(lon, lat) + ... + c_{N,i}p_N(lon, lat),\tag{2}$$

where $V_i$ is the i:th spatial pattern obtained by PCA analysis of the IDF curves (Equation 1), $c_{0,i}$ is the intercept and $c_{1,i}$, $c_{2,i}$, ..., $c_{N,i}$ are the coefficients associated with the predictors $p_1$, $p_2$, ..., $p_N$ for principal component i $\in$ [1,2].

The estimated principal components were then combined with the corresponding eigenvectors and eigenvalues:

$$\hat{X} = U_{1,2}\Lambda_{1,2}\hat{V}_{1,2}^T,\tag{3}$$

where the matrix $\hat{X}$ contains the estimated IDF curves, $\hat{V}_{1,2}$ is a matrix with the estimated eigenvectors $\hat{V}_1$ and $\hat{V}_2$ (Equation 2),

and $U_{1,2}$ and $\Lambda_{1,2}$ are matrices holding the first two leading eigenvectors and eigenvalues, respectively (a subset of $U$ and $\Lambda$ from Equation 1).

Model fitting was performed by Bayesian linear regression, using the R-package 'BAS' (see Clyde et al., 2011, 2018, Chapters 6-8). A Markov Chain Monte Carlo (MCMC) resampling method was used for stochastic exploration of the model space. The number of predictors was reduced by applying the Median Probability Model (MPM) rule: including only variables

with a marginal posterior inclusion probability (pip) of at least 0.5 (Barbieri and Berger, 2004).

## 2.4 Evaluation of results

### 2.4.1 Cross-validation

To evaluate model skill, a leave-one-out cross-validation was performed in which the predictand and predictor data of one station were excluded from model calibration. The statistical model was subsequently applied to the climatological and geo-

graphical data of the excluded station to estimate return values. This procedure was repeated so that independent estimates of the return values were obtained for all stations. The root-mean-square error (RMSE) and relative RMSE between the original return values and independent estimates obtained through cross-validation were then calculated as described in Appendix A.





### 2.4.2 Confidence intervals

Confidence intervals of the estimated return values, $\hat{X}$, were calculated based on the standard errors of the estimated PCs,
$\hat{V}_1$ and $\hat{V}_2$, which were provided as output of the Bayesian linear regression function. Since the principal components are
orthogonal, the error propagation equation for linear combinations (Ku, 1966) simplifies to:

$$\sigma_{\hat{X}} = \sqrt{\sum_i^2 \left(\frac{\partial X}{\partial V_i}\right)^2 \sigma_{V_i}^2} = \sqrt{\sum_i^2 (\Lambda_i U_i)^2 \sigma_{V_i}^2} \tag{4}$$

where $\sigma_{\hat{X}}$ is the total error of the estimated return values $\hat{X}$, $\sigma_{V_i}$ is the standard errors of the i:th principal component, $V_i$, and
$U_i$ and $\Lambda_i$ are the i:th eigenvector and eigenvalue, respectively (see Equations 1 and 3).

### 2.5 Comparison with other methods

The IDF curves estimated with Bayesian inference of principal components, as described above, were compared with the
simple approximate formula for estimating return values derived by Benestad et al. (2021):

$$X_L = \alpha\mu \left(\frac{L}{24}\right)^\zeta \ln f_w \tau, \tag{5}$$

where $X_L$ is the return value (unit: mm) for the duration $L$ (unit: hours), $\mu$ (unit: mm/day) and $f_w$ (unit: fraction of days) are
the wet-day mean precipitation and wet-day frequency, respectively, calculated from daily precipitation observations, and $\alpha$
and $\zeta$ are empirical correction factors that vary with the return period $\tau$ (unit: years). The value of $\alpha$ varies linearly with the
logarithm of the return period according to $\alpha = 1.256 + 0.064 \ln(\tau)$. Values of $\zeta$ have been fitted for a range of return periods
(0.4251593, 0.4185929, 0.4161947, 0.4147515, 0.4144257, 0.4137387, 0.4134449, 0.4134594 for 2, 5, 10, 20, 25, 50, 100, 200
year return periods) and are obtained by interpolation for other values of $\tau$. Equation 5 has been implemented in the R-package
'esd' (Benestad et al., 2015) where it is available as the function `day2IDF()`.

## 3 Results

### 3.1 PCA analysis of the IDF curves

As mentioned earlier, Principal Component Analysis was applied to the IDF statistics as described in Section 2.2 with the
purpose of reducing the dimensionality of the data. The five leading principal components explained 74 %, 9 %, 3 %, 3 %,
and 2 % of the variability of the IDF statistics, respectively. At least two PCs need to be retained in order to explain 80% of
the variance or five PCs for the cumulative explained variance to exceed 90%. In the Scree diagram (Figure 3), an elbow was
identified between $V_2$ and $V_3$, suggesting that the two leading PCs represent the most relevant information. However, based on
the comparison between the IDF shapes $U_i$ and the original return values, statistically significant correlations (with $p < 0.01$)
were found for i $\in$ [1,4], suggesting that the leading four PCs all hold information of some importance.





The spatial pattern associated with the first principal component, $V_1$, was characterised by a west-east and south-north gradient, with a strong contrast between the western and southern coasts of Norway on the one hand, and the inland and mid-to-northern Norway on the other hand (not shown here, see Figure S8 in the Supplementary material). The values of $V_1$ were of the same sign at all stations, meaning that it described a pattern with positive correlations among all stations. The second principal component, $V_2$, ranged both positive and negative values and displayed a gradient from the inland region, where

$V_2 < 0$ at most stations, to the western and southern coast, where $V_2 > 0$. The higher-order spatial patterns had less coherent spatial structures, also spanning positive and negative values.

A reconstruction of the IDF statistics (Figure S9 in the Supplementary material) from only the first PC showed that it determined the basic slope and level of the IDF curves. The second PC altered the curvature: in the stations where $V_2 > 0$, the second PC made the IDF curve more convex, i.e. decreased return values for short to intermediate durations and increased them

for long durations. In stations where $V_2 < 0$, $PC_2$ had the opposite influence, making the curve more flat or concave instead. Higher-order PCs had little visible influence on the IDF curves and only slightly tweaked their shapes.

### 3.2   Statistical modelling

Statistical models were fitted for the first two principal components of the IDF statistics, $V_1$ and $V_2$, using Bayesian linear regression as described in Section 2.3. The posterior inclusion probability of the coefficients (Table 1) suggested that the

most important predictors for $V_1$ were $\mu_{warm}$ and $d_{ocean}$, and for $V_2$, $\mu_{cold}$ and $t2m_{JJA}$ (Table 1; Figures S10-S11 in the Supplementary material). An attempt was made to fit a model for $V_3$, but no predictors could be identified that gave a better prediction than a model consisting of only the intercept (Figure S12). As no model could be found for $V_3$ and previous analysis indicated that it had a limited influence on the IDF shapes, it was excluded from further statistical modelling and analysis. The models for $V_1$ and $V_2$ were refitted with the selected predictors to keep the size of the predictor set small and reduce the risk of

over-fitting (Wilks, 1995).

Figure 4 demonstrates the effect of the predictor variables on the estimated 200-year return values for station Hamar II. Similar results were seen at other stations and for other return periods. The two predictors included in the model of $V_1$ ($\mu_{warm}$ and $d_{ocean}$) had the most notable influence on the basic shape and level of the IDF curves. An increase in $\mu_{warm}$ or decrease in $d_{ocean}$ gave an overall increase in return values, but more for long durations than short durations, hence increasing the

slope of the IDF curves (Figures 4a and b). A decrease in $\mu_{warm}$ or increase in $d_{ocean}$ had the opposite effect, decreasing return levels and the slope of the curves. The predictor variables that were involved in the model of $V_2$ ($\mu_{cold}$, and $t2m_{JJA}$) altered the curvature of the estimated IDF curves. An increase in $\mu_{cold}$ or decrease in $t2m_{JJA}$ lowered return values for low to intermediate durations (< 6 hours) and increased return values for long durations (> 6 hours), resulting in a more concave upwards curve (Figures 4c and d). A decrease in $\mu_{cold}$ or increase in $t2m_{JJA}$ had the opposite effect, resulting in a more

convex downward curve.

A new set of IDF curves was constructed by combining the predicted principal components $\hat{V}_1$ and $\hat{V}_2$ with the corresponding eigenvectors and eigenvalues (Equation 3). Cross-validation (Section 2.4.1) showed that the estimated return values were robust in the sense that they were very similar whether a station was included in model fitting or not: Comparing return values





estimated by models tuned with all data and by cross-validation, the RMSE was only 0.9 mm, or 3 % in relative terms (see

examples in Figure S14 in the Supplementary material). At the eight example stations, the confidence intervals of the original
IDF statistics and of the estimated return values overlapped for all durations and return periods (Figure 5). The deviation
between estimated and original return values was generally larger in stations with more steep and curved slopes (i.e. large
differences between short and long durations as in panels c, d, and e in Figure 5) compared to the stations with flatter IDF
curves.

A comparison between the original return values and values estimated from the two leading original PCs (Figures 6a and b)
showed that there was some loss of information from discarding higher-order modes of variability, but the RMSE was rather
low (2.3 mm or in relative terms, 9 %) and the PCA analysis added no bias. The Bayesian statistical modelling of the two
leading PCs (Figure 6b) added uncertainty to the return value estimates (RMSE = 5.9 mm, 23 %) but was still more precise
than the simple equation by Benestad (RMSE = 9 mm; 35 %) (Figure 6c; Figure S17). There was no obvious spatial pattern in

the RMSE of the return values obtained by the Bayesian approach (not shown here, see Figure S19). For the simple formula,
the largest biases occurred for short return periods, for which there was a tendency to overestimate the return values, while there
was little bias for longer return periods. For the Bayesian modelling of the first two PCs, there was a tendency to underestimate
high return values and the bias was similar for all return periods. The Bayesian modelling approach was notably better for low
to medium return values ($< 100mm$) but the discrepancies associated with the two methods were of similar magnitude for

higher return values. The improved representation of the IDF statistics by the Bayesian modelling was not surprising since it
involved an optimisation process that found the best slope and curvature of the IDF shapes, whereas the estimates based on
the simple formula only used the two parameters $\mu$ and $f_w$. In the simple formula, the shapes of the IDF curves were fixed in
terms of a fractal describing inter-scalar dependencies. Nevertheless, the two different strategies gave somewhat similar results,
albeit with substantial scatter.

New IDF curves were generated by applying the statistical models to meteorological and geographical data for 240 stations
in Norway, including Svalbard and Jan Mayen (Figure 7). Many of these stations lack long time series of high-quality sub-daily
precipitation data, and thus IDF curves cannot be calculated by ordinary measures.

Comparing the original return values (Figures 1 and S2 in the Supplementary material) to the estimated values (Figures 7
and S21), the estimated IDF curves are smoother than the original curves. There are obvious regional differences in the shapes

of the IDF curves: Towards the west coast, the IDF curves reach the highest values and the curvature tends to be strong (i.e.,
a large difference between the intensity of short and long durations) compared to stations inland and up north. At northern
locations, the IDF curves are lower with a moderate slope. These regional differences were more emphasised in the estimated
return values. Most notably, in the southeast region (south of 62 $^{o}$N, east of 9 $^{o}$E), the range of estimated return values was
considerably more narrow compared to the original IDF data, even though the estimated values represent more stations covering

a larger area.





## 4   Discussion

The regression results presented in Table 1 and Figure 4 indicated that the mean precipitation intensity in the warm season is connected to increased intensity over all timescales examined, whereas a similar increase in the cold season suggested reduced short-term intensity but increased long-term intensity. A similar increase over all timescales in the warm season is consistent

with Equation 5 ($\delta x_\tau = \alpha(L/24)^\zeta \ln(f_w\tau)\delta\mu$) assuming a constant value for $\zeta$, but the deviation from this in winter may suggest that $\zeta$ is not necessarily constant. The difference in seasonal response is likely related to the dominance of different precipitation-generating processes: stratiform in winter and convective in summer. It could also possibly be connected to the warm and cold initiation of precipitation (Rogers and Yau, 1989). The locations with intense winter precipitation, which tend to have high return values for long durations and relatively low return values for shorter durations, are found along the west

coast (Figures 1 and 7, Figures S2 and S24 of the Supplementary material). The climate of this region is characterised by the North Atlantic storm tracks bringing a steady stream of cyclones that often meet land here in fall and winter, giving rise to long and heavy precipitation events, and a relatively cold climate in summer which is not conducive to convection. Further inland, the heavy precipitation events often occur in summer as a result of convection, which is in line with the relatively high return values for shorter durations. Furthermore, there is a degree of orographically forced precipitation along the mountain ranges.

The influence of the distance to the ocean and temperature in summer (Figure 4) also supports this picture, with the curvature of the IDF curves decreasing with increasing $t2m_{JJA}$ and $d_{ocean}$.

An advantage of the proposed method, applying PCA to the IDF data, is that all durations and return periods are considered together. This is not only computationally efficient but also reduces the influence of uncertain or erroneous individual return levels. There are also some potential pitfalls with this approach. First of all, only the first two PCs could be modeled which

could have too much of a smoothing effect on the IDF curves. There could also be nonlinear effects that were not captured by the linear models which could result in underestimated variations in the estimated PCs. The quality of the estimated return levels was limited by the quality of the IDF data that they were based on. A different set of IDF statistics would likely result in statistical models with similar predictors and coefficients, but the PCA and ultimately the estimated IDF curves would be defined by the shape of the IDF data.

IDF statistics tend to have large uncertainties attached to them, as illustrated by the confidence intervals in Figure 5. It can therefore be difficult to evaluate what constitutes a skillful estimation of return values. Although the IDF statistics used in this paper (Lutz et al., 2020) are referred to as the "original" return values, they too were estimated and not directly observed. Is it enough that the confidence intervals of the two estimates overlap? If so, the regionalisation approach presented here is "good enough". On the other hand, the confidence intervals of the PCA-based estimates are for sure underestimated because they

represent only the distribution of the Bayesian regression coefficients, without taking into account the uncertainties of the IDF statistics or climatological information that went into the modelling.

A more direct evaluation of model skill might be between the original return values (that were also obtained as median values of a distribution, see Lutz et al. (2020)) and the confidence intervals of the estimated return values. Using this metric, the statistical modeling was successful at most but not all stations. For the 24-hour duration and 200-year return period, which





is where the largest discrepancies occurred (Figure 6b), there were five stations at which the original return values were outside of the confidence interval of the estimated return values (Figure S15 in the Supplementary material): four stations along the southern coast of Norway, at which the return values were underestimated compared to the original values, and one station on the west coast where the return value is overestimated instead. A closer examination revealed that at two of the stations where the return values were underestimated (Grimstad - Hia and Time - Lye), the problem could be traced to the discarding of

higher-order PCs, as PC4 had a notable influence on the shape of the IDF curves (not shown). At the other stations with large discrepancies, the explanation was not as easy to find but likely connected to the Bayesian regression.

Given that the IDF statistics in Norway are estimated for each duration separately and often based on short time series (Lutz et al., 2020), the smoother shapes of the IDF curves estimated by PCA-based Bayesian regression may, at some stations, be more representative of the precipitation climate than the original return values which they are based on. Hence, the PCA-based

approach with Bayesian inference can be regarded as another 'tool' together with the traditional approach for estimating IDFs and the simple formula proposed by Benestad et al. (2021). They are based on different assumptions, have different strengths and weaknesses, and together they can capture salient information and smooth over errors from single sites, as the different approaches to predicting IDF curves are based on independent methods, e.g. modelling the return values Dyrrdal et al. (2015), estimating future IDF curves based on changes in the precipitation intensity Zhu et al. (2012), and downscaling rainfall intensity

with respect to timescale Rodríguez et al. (2014); Benestad et al. (2021).

As an alternative to PCA, we tried using a weighted set of polynomials to represent the shapes of the IDF curves, fitting return values against time intervals (Figure S6 in the Supplementary material). This method was not successful. First and second-order polynomials were not a good fit and higher-order polynomials, while fitting the data reasonably well for the time intervals for which return value data were available, had wiggly shapes with local minima and maxima that did not make sense

as IDF curves.

One interesting question is how to use the IDF modelling approach presented here in the context of climate change projections. For the case of Norway, the key variables expected to change are $\mu_{warm}$, $\mu_{cold}$, and $t2m_{JJA}$. By downscaling them, statistically or dynamically, we may be able to infer changes in the IDF through predicting new values for the PCs, assuming that their shapes will be valid in a future climate and the calibrated dependency holds. One limitation is our ability to get reli-

able estimates for the wet-day mean precipitation in the future, which can be challenging in the case of empirical-statistically downscaling. We can also use this approach to provide hypothetical IDF curves for stress-testing (Benestad et al., 2019a). Other caveats may be that these results only apply to Norway and that similar analyses for different regions may find that different factors are important for the shape of the IDF curves. We expect that the shapes of the IDF curves depend on physical aspects such as mesoscale convection, synoptic frontal systems and cyclones, orographic precipitation, and atmospheric rivers and that

they will be sensitive to changes in their occurrence relative to each other.



## 5 Conclusions

We obtained predictions of the shape of IDF curves in Norway with Bayesian inference applied to a PCA representation of the IDF data and conclude that it provides a useful strategy that can be utilised for regionalisation and downscaling of future climate projections.

*Code and data availability.* The data code used to carry out this analysis is available in an R-markdown document provided in the Supplementary material. The daily precipitation and temperature data used in this paper are publicly available and the IDF statistics are available on request from the authors.

### Appendix A: Skill statistics

Statistical measures of the discrepancy between original return values x and fitted values y were calculated as follows:

$$RMSE = \frac{1}{N} \sum_{i=1}^{N} \sqrt{(y_i - x_i)^2} \tag{A1}$$

$$RMSE_{rel} = \frac{RMSE}{\bar{x}} \tag{A2}$$

*Author contributions.* Kajsa Parding and Rasmus Benestad contributed to the data analysis. Julia Lutz supplied the IDF data. All authors participated in writing the manuscript.

*Competing interests.* No competing interests are present.


*Acknowledgements.* This work was supported by the Norwegian Meteorological Institute and the KlimaDigital project (Norwegian Research Council grant: 281059).





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





**Table 1.** Marginal posterior inclusion probability (pip) of the coefficients for different predictor variables in the statistical models for the three leading principal components of the IDF curves, $V_1$, $V_2$ and $V_3$. A pip higher than 0.5 (shown in bold font) indicates that the coefficient is included in the Median Probability Model (MPM), which is the model selection approach used in this study. The included predictor variables are the wet-day mean precipitation in the warm and cold season ($\mu_{warm}$, $\mu_{cold}$), the wet-day frequency in the warm and cold season ($f_{w_{warm}}$, $f_{w_{cold}}$), the mean summer temperature ($t2m_{JJA}$), and the latitude, altitude and distance to the ocean ($d_{ocean}$).

| Predictor | pip $V_1$ | pip $V_2$ | pip $V_3$ |
|---|---|---|---|
| Intercept | **1.0000** | **1.0000** | **1.0000** |
| $\mu_{warm}$ | **1.0000** | 0.1089 | 0.0220 |
| $\mu_{cold}$ | 0.1638 | **1.0000** | 0.0221 |
| $f_{w_{warm}}$ | 0.1114 | 0.0830 | 0.0617 |
| $f_{w_{cold}}$ | 0.1533 | 0.0796 | 0.0870 |
| $t2m_{JJA}$ | 0.1621 | **0.9756** | 0.0285 |
| latitude | 0.2854 | 0.0815 | 0.0223 |
| altitude | 0.1225 | 0.1108 | 0.0252 |
| $d_{ocean}$ | **0.8426** | 0.0850 | 0.0542 |

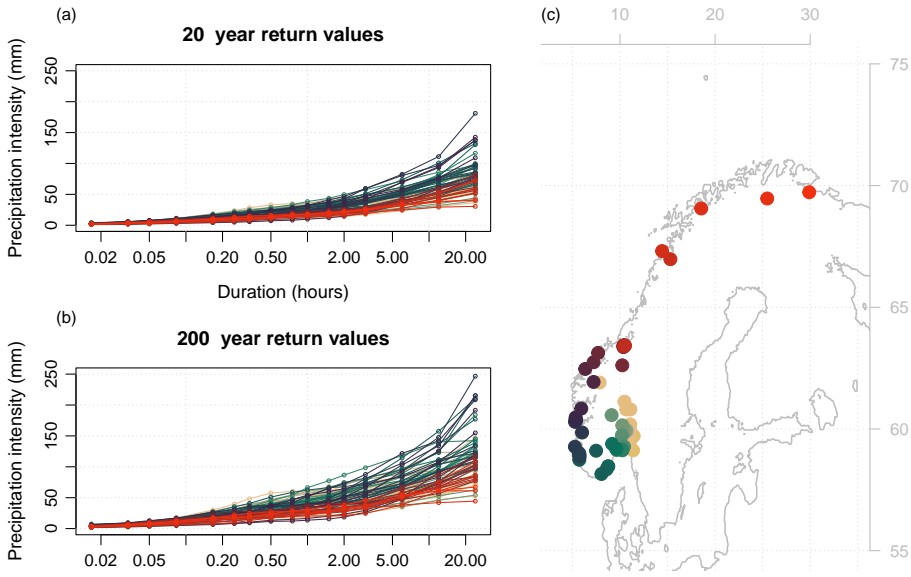

**Figure 1.** Return values for 74 Norwegian stations for the (a) 20 and (b) 200-year return periods. The colours represent different locations shown in the map (c).



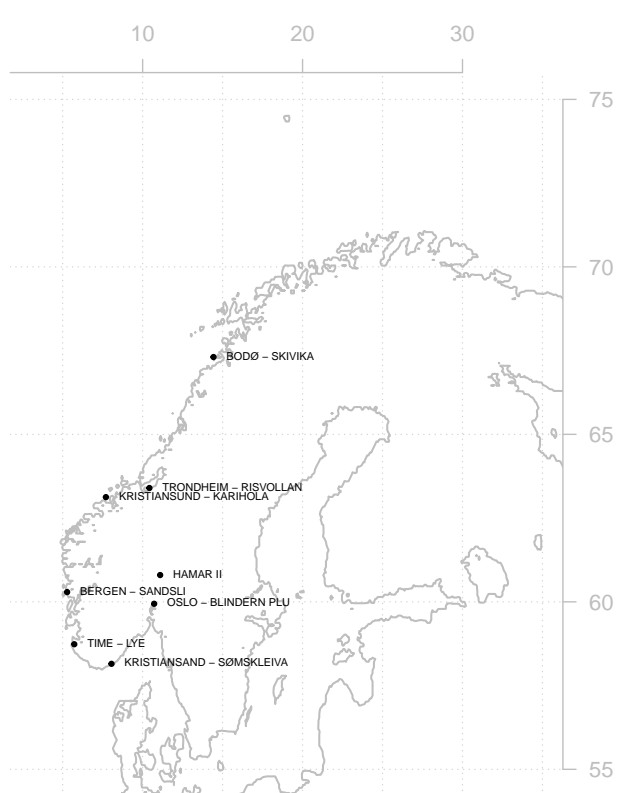

**Figure 2.** Map showing the location of the example stations for visualisation in this paper.

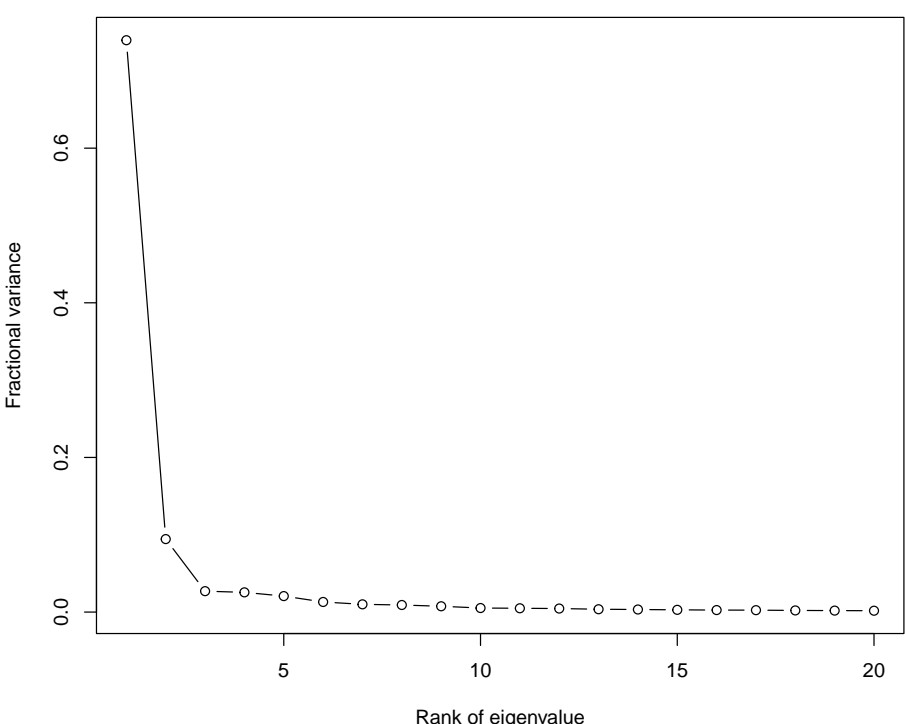

**Figure 3.** Scree diagram of the principal components of the IDF statistics. The figure shows the rank of the eigenvalues (abscissa) plotted against the explained variance (ordinate).



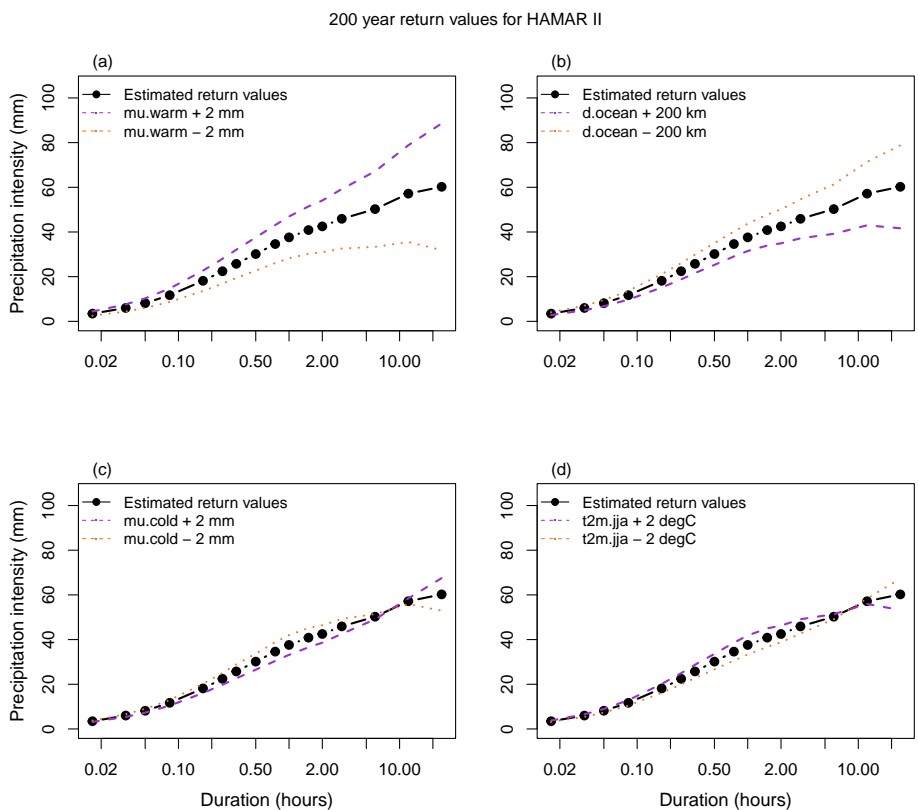

**Figure 4.** Demonstration of the effect of the model parameters (a) $\mu_{warm}$, (b) $d_{ocean}$, (c) $\mu_{cold}$, and (d) $t2m_{JJA}$ on the estimated 200-year return values for the station Hamar II. The plots show return values estimated with parameter values from observations (black lines and dots), as well as with increased (purple dashed lines) and decreased (orange dotted lines) parameter values.



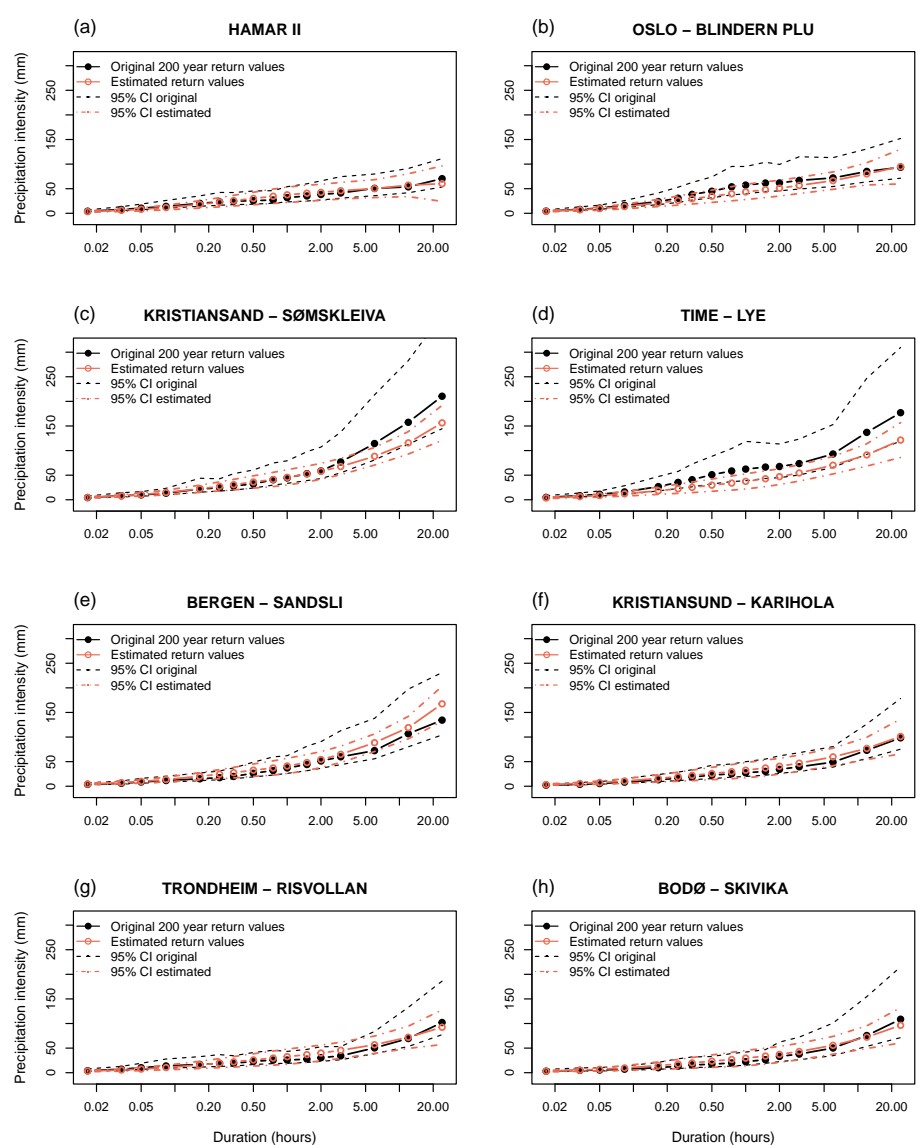

**Figure 5.** Estimated 200-year return values for eight example stations: (a) Hamar II, (b) Oslo - Blindern Plu, (c) Kristiansand - Sømskleiva, (d) Time - Lye, (e) Bergen - Sandsli, (f) Kristiansand - Karihola. and (g) Bodø - Skivika. The plots show the original IDF curves (black) as well as return values estimated by Bayesian inference as described in this paper (coral). Dashed lines show the confidence interval (2 standard errors) of the original (dashed black) and estimated (dashed coral) IDF curves.

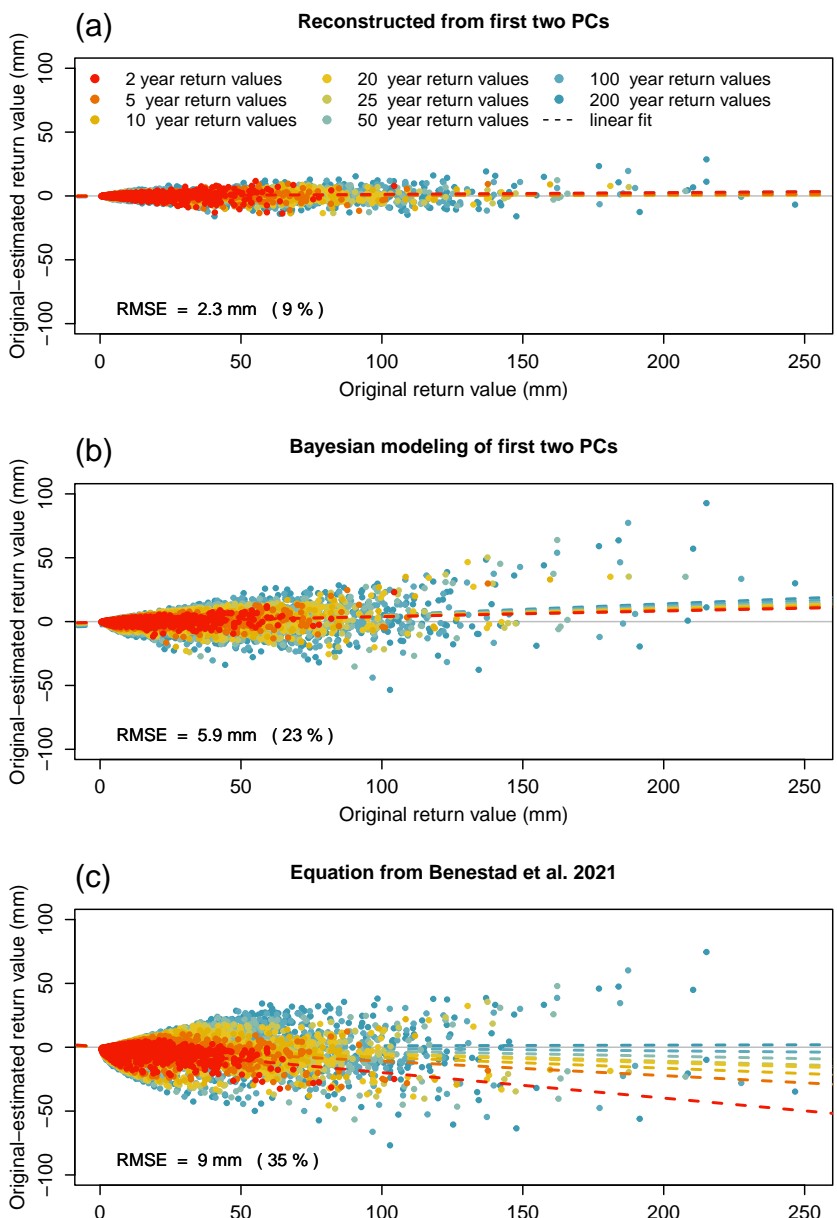

**Figure 6.** Original return values plotted against the error of the estimated return values (original - estimated) calculated (a) from the two leading PCs of the original return values, (b) by Bayesian modelling of the two leading PCs of the return values, and (c) using the equation presented in Benestad et al. (2021). The colours represent different return periods (see legend in panel a), points show individual return values and dashed lines show linear regressions for each return period. The RMSE and relative RMSE of the estimated and original return values are displayed in the lower left corner of each panel.

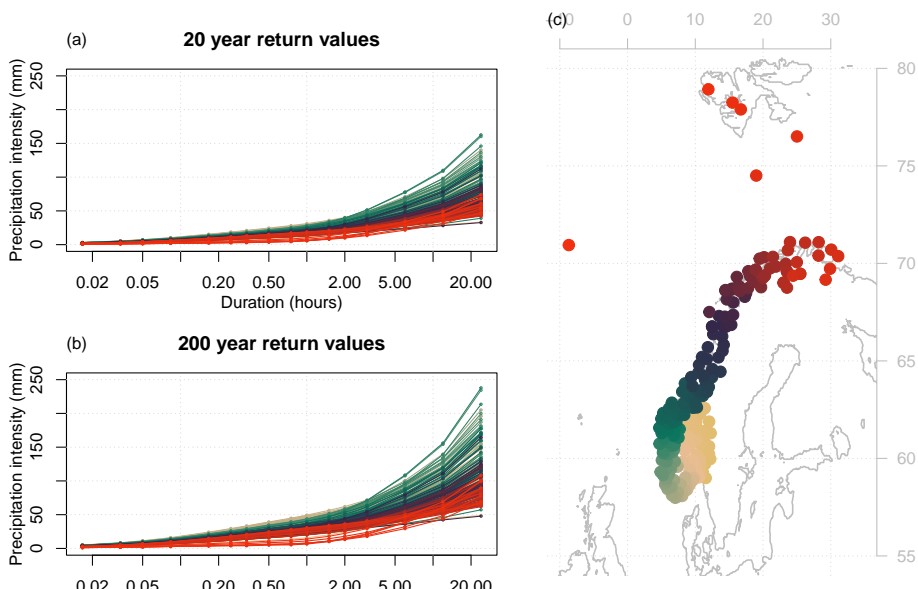

**Figure 7.** Estimated return values for 240 Norwegian stations for the (a) 20-year and (b) 200-year return period, calculated using statistical models applied to temperature and precipitation data that were not used for model calibration. The map (c) shows which locations the colours in the return value plots represent.