# Peer review of "A principal component based strategy for regionalisation of precipitation intensity-duration-frequency (IDF) statistics"

_Hydrology and Earth System Sciences, 2022_

## Author Response (AR1)

HESS-2022-233 | Research article
Submitted on 17 Jun 2022

**A principal component based strategy for regionalisation of precipitation intensity-duration-frequency (IDF) statistics**

Kajsa Maria Parding, Rasmus Emil Benestad, Anita Verpe Dyrrdal, and Julia Lutz

**Response to reviewers**

Anonymous referee #1

This study proposes an empirical statistical modelling approach to estimate the precipitation intensity-duration-frequency (IDF) statistics from 74 stations in Norway. Principal Component Analysis (PCA) was first used to reduce the dimensions and complexity of the IDF of all stations into two sets of principal components (PCs) through the Singular Value Decomposition (SVD). The PCs were then regressed against climatological and geographical information using Bayesian linear regression.

General Comments:

This manuscript is well-written with a good structure, and the proposed methodology is mathematically sound and well-established, which provides a good contribution in statistical inference methods of the IDF curves. Having said that, there are several issues needed to be addressed to better discuss the uncertainty and to enhance the applicability of the method.

☐ The nature and probabilistic behaviour of extreme climate phenomena is known to be influenced by anthropogenic climate change over time, which challenges the fundamental stationary risk concepts in calculating the IDF curves. As a result, the static stationary-based IDF curves may underestimate the occurrence probability of extreme precipitation. I wonder how the proposed methodology address the non-stationary issue (e.g. historical trends in the probability of heavy rainfall) and incorporate the influence of non-stationary conditions on IDF curves?

*The proposed methodology does not address the non-stationarity issue. Both the IDFs (estimated from sub-daily precipitation data) and the climatological information (from daily climate data) that were used in the study are based on data from stations that meet some requirements of data availability, but not necessarily from the exact same period. Requiring a common period would have reduced the number of stations as well as the sample size for the stations that were included, which would also have reduced the representativeness.*

*A common way to take climate change into consideration in IDF statistics is to simply multiply the return values by a climate change factor that is derived from*

*climate modeling. Another way to achieve this would be to use a statistical inference method like the one described in this study and apply it to downscaled climatological data for the future. Such an approach may add value to the IDF estimates by being site specific and scaling differently to different durations and return periods. (But as noted by both reviewers, we have not applied our method to downscaled climate data for the future yet, so this is hypothetical.)*

☐ The Benestad et al, (2021) simple approximate formula that used to compare and assess the proposed methodology in this study is only one of many approaches in estimating the return values and predicting the IDF curves, and it is based on some assumptions (e.g. deliberate choice of using L=24) and is calibrated only from Oslo and validated at some independent sites in Norway. The formula has not been, to my knowledge, vigorously tested worldwide, and thus, remains regional specific. To ensure the robustness of the proposed methodology and increase its applicability, I suggest the authors evaluate the proposed methodology with several more commonly used statistical inference methods (e.g. parametric formulation of IDF relationships based on Koutsoyiannis et al. (1998) framework, distribution fitting using L-moments/probability weighted moments estimation, and regionalization methods such as the Index Flood method). In this way, the readers will have better ideas on how the proposed methodology performs against those widely used methods.

- Koutsoyiannis, D., Kozonis, D., & Manetas, A. (1998). A mathematical framework for studying rainfall intensity-duration-frequency relationships. Journal of Hydrology, 206(1-2), 118-135. https://doi.org/10.1016/S0022-1694(98)00097-3.

*We compare our results to two other sets of return values, the ones calculated using the Lutz et al. 2020 method, which are the IDF statistics that we use as a sort of "truth" when calculating the RMSE, and the simple relationship developed by Benestad et al. (2021). The Benestad method has not been extensively tested and we think that is a good reason to include it here, not as a gold standard but as another regionalization method that also needs to be evaluated. It would be interesting to compare with other methods as well, but we consider it beyond the scope of this paper.*

☐ As the authors correctly stated, the accuracy of IDF curves depends on the quality of input data and the statistical inference methods. While this study focuses on the latter, the former should not be neglected and a better discussion in this regard could be done. Since considerable amount of research (e.g. Eldardiry et al., 2015; Marra et al., 2017; Degaetano & Castellano, 2017) have been done on investigating the use of alternative sources of rainfall measurements (e.g. radar, satellite-based precipitation, downscaled global/regional climate models' precipitation simulations, and reanalysis products) in constructing the IDF curves, I wonder how these alternative sources of

information could potentially be used and supplement with the ground stations in the study region?

- Eldardiry, H., Habib, E., & Zhang, Y. (2015). On the use of radar-based quantitative precipitation estimates for precipitation frequency analysis. Journal of Hydrology,531, 441–453. https://doi.org/10.1016/j.jhydrol.2015.05.016.
- Marra, F., Morin, E., Peleg, N., Mei, Y., & Anagnostou, E. N.: Intensity–duration–frequency curves from remote sensing rainfall estimates: comparing satellite and weather radar over the eastern Mediterranean, Hydrol. Earth Syst. Sci., 21, 2389–2404, https://doi.org/10.5194/hess-21-2389-2017, 2017.
- DeGaetano, A. T., & Castellano, C. M. (2017). Future projections of extreme precipitation intensity-duration-frequency curves for climate adaptation planning in New York State.Climate Services,5,23–35. https://doi.org/10.1016/j.cliser.2017.03.003.

*Other sources of information could definitely be a useful complement to the surface based observations. While daily temperature and precipitation observations are more widely available than sub-daily data, there are issues with missing data in many regions. Gridded products have their limitations when it comes to representing extreme precipitation, as there is an inherent difference between point observations and the spatial average of a grid point. This is especially notable in a country such as Norway with dramatic topography and large climatological variations on small scales (high peaks, deep valleys and fjords), where precipitation may differ considerably within a grid cell. Nevertheless, data from remote sensing could serve as an additional source of information.*

Specific Comments:

- ☐ L34-35: Certainly building the relationship between IDF curves and some climatological and geographical factors is one way to regionalize the IDF curves, but there are also other ways to estimate the IDF curves such as using radar and remote sensing data (e.g. Eldardiry et al., 2015; Marra et al., 2017; Ombadi et al., 2018; Sun et al., 2019). It would be appreciated if the authors could add some discussions in this regard.
  - Ombadi, M., Nguyen, P., Sorooshian, S., & Hsu, K. L. (2018). Developing intensity-duration-frequency (IDF) curves from satellite-based precipitation: Methodology and evaluation. Water Resources Research, 54(10), 7752-7766. https://doi.org/10.1029/2018WR022929.
  - Sun, Y., Wendi, D., Kim, D. E., & Liong, S. Y. (2019). Deriving intensity–duration–frequency (IDF) curves using downscaled in situ rainfall assimilated with remote sensing data. Geoscience Letters, 6(1), 1-12. https://doi.org/10.1186/s40562-019-0147-x.

*A discussion about and references to studies of IDF regionalization using remote sensing data have been added to the manuscript.*

☐ L40-41: It is not clear that why calculating IDF curves for each grid is impractical and computationally expensive as research has been done at the global scale, i.e. Courty et al., 2019. Please clarify.

- Courty, L. G., Wilby, R. L., Hillier, J. K., & Slater, L. J. (2019). Intensity-duration-frequency curves at the global scale. Environmental Research Letters, 14(8), 084045.

*Calculating IDF curves for each point in a grid tends to be more computationally expensive than doing the same for a set of stations, simply because it involves more data. Of course, it is not so impractical and expensive that it cannot be done and it can definitely be worth doing to get a more complete spatial coverage. However, for Norway and other mountainous and coastal regions, the resolution of the grid would have to be very high. One example is the regionalization done by Dyrrdal et al. (2015) who estimated IDF curves for Norway based on the seNorge data set which has a 1x1 km resolution.*

*The sentence has been changed to clarify this.*

*Dyrrdal, A., Skaugen, T., Stordal, F. & Førland, Eirik. (2014). Estimating extreme areal precipitation in Norway from a gridded dataset. Hydrological Sciences Journal. 61. 141217125340005. 10.1080/02626667.2014.947289.*

☐ L73-74: It seems to me that there are no analysis done on using the proposed statistical modelling in combination with future projections of meteorological quantities. I could have missed the material, please correct me if I am wrong. If not, please show the results or the objective is over-stated otherwise.

*No, you are right, this has not been done yet. We have adjusted the text to clarify that this is a plan for the future rather than an analysis that has been performed.*

☐ L88-90: Can the authors comment on the quality of the data (e.g. % of missing values) please?

*For the IDF curves that were calculated and provided by Dr. Julia Lutz, the stations that were included had to have at least 80% data availability per April - October season and a minimum of 10 seasons that met this requirement. For the daily temperature and precipitation data used in this study, we had a similar requirement of data availability (10 years of available data). The observational data, both sub-daily and daily precipitation and temperature, has also undergone a quality assessment before being made publicly available by the Norwegian Meteorological Institute.*

*We added an investigation of the data availability of daily precipitation and temperature data at the stations that were included in the study in the supplementary material (Figure S23). Temperature and precipitation data are*

*available for different periods at different locations, and in some instances with no overlap in time. Few stations have long complete observational records of both temperature and precipitation. For example, only four of the selected stations have a data availability of at least 70% in the period 1970-2020 for both temperature and precipitation. Since we do not have any restrictions on the specific period for which observations should be available and have no information about the precise period of the data that went into the IDF computations, there is no guarantee that the IDF curves and the climatological data that were used to tune the regionalization models represent the same period. This is a weakness of the study, and an unfortunate result of the sparseness of historical observations. Other sources of information, such as the gridded seNorge data set, could be used to alleviate this problem. A discussion of the data availability and quality has been added to the manuscript.*

☐ L90-92: Is it possible to have the same station with temperature data assigned to two different IDF stations, given the sparse spatial coverage of the network?

*Yes, it is possible and it happens. Out of the 74 stations considered in this study, 26 are assigned to multiple stations and 48 are associated with only one precipitation station. The multiple assignment occurs primarily in the more densely populated parts of the country where there are several precipitation stations in a relatively small area (around Stavanger, Trondheim and Oslo). Since the temperature is rather spatially homogeneous, this is likely not a very large issue when looking at climatological values. However, it would be interesting to look further into this, as a part of a larger investigation into the influence of data quality and availability on the estimated IDF curves.*

☐ L196-205: The sensitivity of the IDF curves on the predictors were examined by holding one variable constant at a time. I wonder how the combined effect of the predictors influences on the shape and level of the IDF curves? It would be appreciated if the authors could do a more in-depth analysis here.

*We did some investigations of how changing combinations of parameters influence the shape of the IDFs, and while they did not reveal anything unexpected, the new figure was helpful in showing how the various parameters can interact. For example, it clearly showed how the wet-day mean precipitation in the warm season (April - September) had a more dominant influence on the estimated IDFs than the wet-day mean in the cold season (October - March). The new plot is included in the Supplementary material as Figure S13.*

Remarks:

☐ Figure 2: the IDF curves for all return periods for eight stations (Figure S4) could be shown here alongside with the geographical locations of the stations, i.e. combining Figure S4 and Figure 2.

*That's a good idea. We have combined the two figures and replaced Figure 2 with the new and improved version.*

In the study, the authors employed the principal component analysis and the Bayesian linear regression method to investigate the precipitation Intensity-duration-frequency (IDF) curves and their spatial distribution in Norway, and also explored the prediction by considering both geographical conditions and local climate characteristics. The description of extreme rainfall events, especially at short timescales, is important for the control and management of natural disasters, but it is also a big challenge. Thus, the new approach proposed in this paper for prediction of precipitation IDFs is useful and could be a good reference for the related studies. Overall, the paper is well written and easily readable. However, the following issues are suggested to be considered for further improving the quality of the paper before its publication.

☐ The first issue is about the stability of the relationship between the shapes of IDFs and those predictors selected. For the geographic predictors, their values are constants, however, the values of those climatic predictors closely depend on the data periods selected. Their values only based on short data period in this study would have big bias from the true values, which would significantly influence the stability of the relationship between the shapes of IDFs and climatic predictors. Especially, the authors discussed in Section 4 that the approach can be used for downscaling of climate change projections, if we cannot ensure the stable relationship between the shapes of IDFs and these predictors, how to do downscaling and ensure the reliability of the results? At least the issue should have a deep discussion including the influence or the uncertainty analysis in Figure 4 and 5.

*Changes have been observed in the heavy precipitation in the Nordic-Baltic region. Dyrrdal et al. (2021) reported positive trends in daily annual precipitation maxima in a majority of stations in the regions, with strong changes in southeast Norway. This is likely to influence the estimation of IDFs. The framework that we are using in this study does not take non-stationarity into account. One issue is whether or not the statistical model that connects the climatological values and the principal components of the IDFs are stationary. This, we have not looked into. Another is whether the climatological values change so much over time that the difference in the periods of data availability from station to station has an influence on the estimated IDFs.*

*We have added an analysis of the trends in the climatological variables of importance in the Supplementary material. The analysis indicates that while there are few stations with significant trends in the wet-day mean in the cold or warm season, many stations display a significant warming in the summer season (Figure S24). However, as demonstrated in Figure 4, it is the changes in*

*precipitation in the warm season that has the strongest influence on the estimated IDFs.*

*We also tried calculating IDF curves based on data from two different periods: 1970-1995 and 1995-2020. For this analysis, we selected 36 stations with long data records of both temperature and precipitation. At a majority of the stations, there was an increase in the estimated return values from the first period to the second. On average, the difference was small, but larger changes occured. The preliminary results of this analysis suggests that the observational period that goes into the climatological values used in the regionalization can have a strong influence on the estimated IDFs. We have added a discussion of these findings in the manuscript.*

*Dyrrdal, A. et al. (2021) Observed changes in heavy daily precipitation over the Nordic-Baltic region, Journal of Hydrology: Regional Studies, 38, 100965, ISSN 2214-5818, https://doi.org/10.1016/j.ejrh.2021.100965.*

☐ The second issue is about the presentation of the results in Figure S8. Considering that the spatial pattern from Figure S8 is an important information for understanding the spatial distribution of IDFs, it is suggested that the Figure and its related information can be added in the main document rather than in the Supplementary material.

***Figure S8 has been added to the main manuscript.***

☐ The third is about the applicability of the new approach proposed. Actually, the authors very briefly mentioned it in the last paragraph in the paper, however, it is not enough. As the study area of Norway has special climatic conditions, how about the applicability of the new approach when applying to other regions with totally different climatic conditions? It is suggested more contents be added to discuss the issue.

*The general methodology could be appropriate in other regions, but the statistical model would have to be trained on a different set of IDF statistics and climatological and geographical data. The coefficients of the statistical model used in this paper are not expected to be universally applicable, and other model parameters may be more appropriate in other regions. For example, other seasonal divisions than the warm/cold seasons used in this study may be more useful to describe the annual cycle of precipitation and processes associated with heavy precipitation in other regions. Other geographical descriptors, such as the altitude or the slope orientation could prove more important than the distance to the ocean. The PCA would also pick up on different large scale patterns, being applied to a different set of IDFs. **A discussion on this topic has been added to the manuscript.***

☐ Besides, how to determine the predictor of "distance to ocean". The key issue can be explained more clearly.

*The distance to the ocean is the shortest distance from a point to the coast line. The function that is used to calculate the distance to the ocean is defined in the appendix (the RMarkdown file) for those who are interested in the details. This has been clarified in the manuscript.*